# VLSM-Ensemble: Ensembling CLIP-based Vision-Language Models for Enhanced Medical Image Segmentation

**Julia Dietlmeier**[1] [iD]          JULIA.DIETLMEIER@INSIGHT-CENTRE.ORG
[1] *Insight Research Ireland Centre for Data Analytics, DCU, Dublin, Ireland*
**Oluwabukola Grace Adegboro**[*2]   OLUWABUKOLA.ADEGBORO2@MAIL.DCU.IE
**Vayangi Ganepola**[*2]          VAYANGI.GANEPOLA2@MAIL.DCU.IE
[2] *Research Ireland Centre for Research Training in Machine Learning, DCU, Dublin, Ireland*

**Claudia Mazo**[3]              CLAUDIA.MAZO@DCU.IE
[3] *School of Computing, Dublin City University, Dublin, Ireland*

**Noel E. O'Connor**[1]          NOEL.OCONNOR@INSIGHT-CENTRE.ORG

**Editors:** Accepted for publication at MIDL 2025

## Abstract

Vision-language models and their adaptations to image segmentation tasks present enormous potential for producing highly accurate and interpretable results. However, implementations based on CLIP and BiomedCLIP are still lagging behind more sophisticated architectures such as CRIS. In this work, instead of focusing on text prompt engineering as is the norm, we attempt to narrow this gap by showing how to ensemble vision-language segmentation models (VLSMs) with a low-complexity CNN. By doing so, we achieve a significant Dice score improvement of 6.3% on the BKAI polyp dataset using the ensembled BiomedCLIPSeg, while other datasets exhibit gains ranging from 1% to 6%. Furthermore, we provide initial results on additional four radiology and non-radiology datasets. We conclude that ensembling works differently across these datasets (from outperforming to underperforming the CRIS model), indicating a topic for future investigation by the community. The code is available at https://github.com/juliadietlmeier/VLSM-Ensemble.
**Keywords:** BiomedCLIP, CLIP, CLIPSeg, Image segmentation, Vision-language models.

## 1. Introduction

Vision-Language Foundation Models (VLFM) such as CLIP (Radford et al., 2021) and BiomedCLIP (Zhang et al., 2024) are inherently multimodal and are pretrained on large figure-caption datasets. Historically, CLIP was one of the first VLFMs that was pretrained on 400 million image-text pairs from the Internet using contrastive learning. BiomedCLIP is a very recent VLFM specifically created to process biomedical data and is pretrained on 15 million image-text pairs extracted from biomedical research articles in PubMed Central. BiomedCLIP and CLIP can be extended to work in image segmentation scenarios, and some corresponding models are BiomedCLIPSeg (Poudel et al., 2024) and CLIPSeg (Lüddecke and Ecker, 2022). These models consist of separate CLIP-based image and text encoders and a combined image-text decoder. Both CLIP encoders and the decoder have visual transformer-based architectures. During fine-tuning, the CLIP encoders remain frozen and only decoder weights are updated.

---

\* Contributed equally

We are inspired by the results frequently achieved in many coding challenges where the winning approaches successfully use ensembling in one form or another (Ferreira et al., 2023). Specifically, we apply one of the ensembling approaches termed *stacking* - a machine learning technique for creating strong models from multiple weak learners.

Our contributions are as follows. In this work, we do not address zero-shot capabilities of VLSMs, but rather focus on joint fine-tuning. We design three ensemble architectures where we combine two VLSMs with a low-complexity UNet-like model: BiomedCLIPSeg with UNet (**BiomedCLIPSeg-A**) and CLIPSeg with UNet (**CLIPSeg-B**). The third architecture is the **Ensemble-C** of BiomedCLIPSeg (Poudel et al., 2024), CLIPSeg (Lüddecke and Ecker, 2022) and UNet (Ronneberger et al., 2015) as shown in Figure 1.

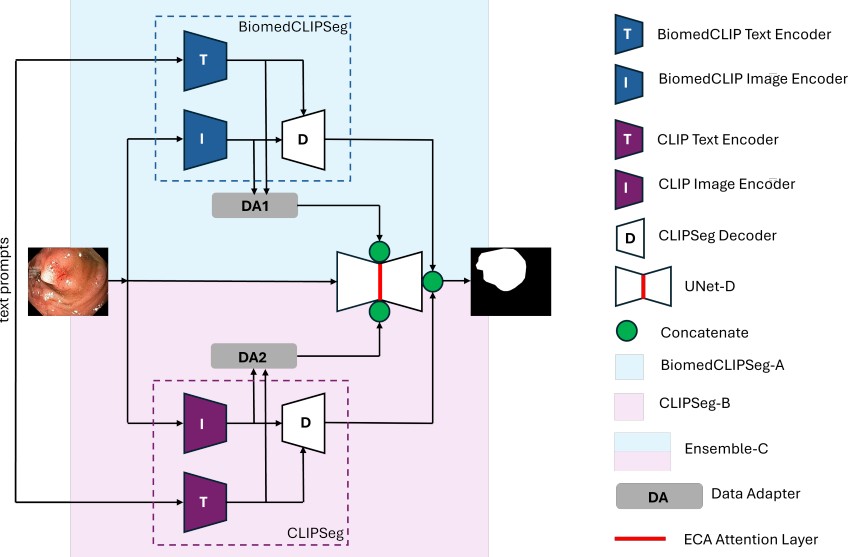

Figure 1: Proposed ensembles of CLIP-based VLSM architectures with the UNet-D.

## 2. Motivation, Methodology and Results

While designing the ensembling pipelines, the main motivation is to leverage the semantically-rich outputs from the text and image encoders of both BiomedCLIP and CLIP and concatenate them with the bottleneck of a basic UNet model with four encoder-decoder blocks (without a pretrained backbone) and skips. For this purpose, we create two data adapters DA1 and DA2, that adapt each VLSM to the tensor shape of UNet's bottleneck layer. We also integrate an Efficient Channel Attention (ECA) layer (Wang et al., 2020).

Our experimental test bench, text prompts, and data splits are based on the work by (Poudel et al., 2024). We design our ensembling models and prepare the data in such a way that they fit into this framework. We implement all models in PyTorch and train all models (except UNet-D) on our system with Nvidia GeForce RTX 2080 Ti GPU with 11GB VRAM. The UNet-D model was trained on Kaggle using the Tesla P100-PCIE-16GB GPU. We finetune with early stopping with patience 20. We use the AdamW optimizer with an initial learning rate of $10^{-5}$ and a combination of binary cross-entropy and Dice losses.

The batch size is 10. We evaluated our models on five public datasets: Kvasir-SEG (Jha et al., 2020), ClinicDB (Bernal et al., 2015), BKAI (Lan et al., 2021; An et al., 2022), BUSI (Al-Dhabyani et al., 2020), and CheXlocalize (Saporta et al., 2022).

Table 1: Dice ↑ score results *averaged over all text prompts. Performance gains $\Delta$ are shown in gray cell colors such as $\Delta_{CLIPSeg-B}$ is computed relative to CLIPSeg-B.

| Model | Kvasir[1] | ClinicDB[1] | BKAI[1] | BUSI | CheXlocalize |
|---|---|---|---|---|---|
| *CRIS (Wang et al., 2022) | 0.83043 | 0.79390 | 0.85122 | 0.65453 | 0.55425 |
| *BiomedCLIPSeg | 0.81457 | 0.75967 | 0.75120 | 0.62548 | **0.56881** |
| *BiomedCLIPSeg-A (ours) | 0.83976 | 0.80635 | 0.81423 | 0.64908 | 0.56634 |
| $\Delta_{BiomedCLIPSeg}$ | 2.519% | 4.668% | 6.303% | 2.36% | -0.247% |
| *CLIPSeg | 0.85744 | 0.77343 | 0.82648 | 0.62271 | 0.54875 |
| *CLIPSeg-B (ours) | 0.86523 | 0.78966 | **0.85234** | 0.63529 | 0.54901 |
| $\Delta_{CLIPSeg}$ | 0.779% | 1.623% | 2.586% | 1.258% | 0.026% |
| *Ensemble-C (ours) | **0.86795** | **0.81166** | 0.84063 | **0.66008** | 0.52786 |
| $\Delta_{CLIPSeg-B}$ | 0.272% | 2.2% | -1.171% | 2.479% | -2.115% |
| UNet-D (ours) | 0.60215 | 0.33580 | 0.48360 | 0.56078 | 0.29179 |

Table 1 shows that individual ensembling significantly improves the performance of the BiomedCLIPSeg and CLIPSeg models, especially for the polyp[1] datasets. It can also be seen that the UNet-D component on its own performs poorly across all datasets.

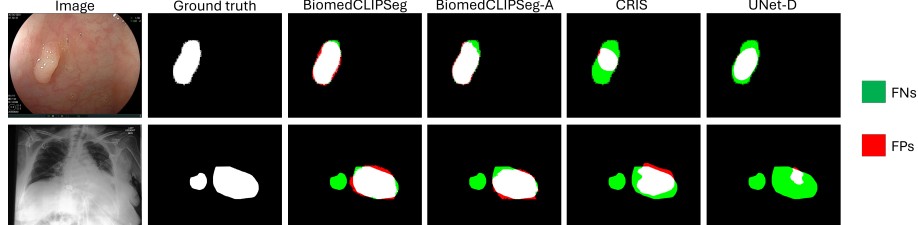

Figure 2: Selected qualitative results for the BKAI (top) and CheXlocalize (bottom).

## 3. Conclusion and Future Directions

We achieved high average Dice gains on polyp[1] datasets with our ensembled BiomedCLIPSeg-A model. The selected qualitative results in Figure 2 show fewer False Positives (FPs) for the BiomedCLIPSeg-A compared to BiomedCLIPSeg and fewer False Negatives (FNs) compared to CRIS. However, neither the VLSM model recovers the small mask on the left, while the CRIS model has the most FNs. In the future, we anticipate understanding the reason for some negative gains, the global and individual relationships between quantitative and qualitative results, and investigating the impact of text prompts on performance.

## Acknowledgments

This work was funded by the Insight Research Ireland Centre for Data Analytics and partly supported by Taighde Éireann – Research Ireland under Grant number 18/CRT/6183 (Adegboro, Ganepola).

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
