# OpenReview forum: "VLSM-Ensemble: Ensembling CLIP-based Vision-Language Models for Enhanced Medical Image Segmentation"
_MIDL.io/2025/Short_Papers — MIDL 2025 - Short Papers_

### Official Review · Reviewer_D9mk · 2025-04-28

**Rating:** 4
**Confidence:** 4

**Summary:**

The paper proposes an ensemble of vision-language models and a low-complexity CNN for segmentation of polyps. Dice scores increased in some cases with the ensemble of models.

**Strengths:**

1. Combination of multiple ensemble approaches for polyp segmentation.
2. Reasonably decent explanation of the model architecture that was used for training.
3. Evaluation on multiple datasets.

**Weaknesses:**

1. No comparison against nnUNet.
2. No distance errors (Hausdorff or normalized surface distance) described. Dice only shows overlap, while distance errors capture errors in topology.
3. No statistical testing (I recommend Wilcoxon signed rank test and/or one-way ANOVA with your reference).
4. Your performance improvement is up to 6.3% (only for BKAI and not 6% as a minimum across datasets), while other datasets show 1-6% improvement. I would rephrase the sentence in the abstract as it is misleading.

---

### Decision · Program_Chairs · 2025-05-01

Accept